# Trust-Based Data Communication in Wireless Body Area Network for Healthcare Applications

**Sangeetha Ramaswamy**  **and Usha Devi Gandhi ***

School of Information Technology and Engineering, Vellore Institute of Technology, Vellore 632014, India
* Correspondence: ushadevi.g@vit.ac.in

**Abstract:** A subset of Wireless Sensor Networks, Wireless Body Area Networks (WBAN) is an emerging technology. WBAN is a collection of tiny pieces of wireless body sensors with small computational capability, communicating short distances using ZigBee or Bluetooth, an application mainly in the healthcare industry like remote patient monitoring. The small piece of sensor monitors health factors like body temperature, pulse rate, ECG, heart rate, etc., and communicates to the base station or central coordinator for aggregation or data computation. The final data is communicated to remote monitoring devices through the internet or cloud service providers. The main challenge for this technology is energy consumption and secure communication within the network and the possibility of attacks executed by malicious nodes, creating problems for the network. This system proposes a suitable trust model for secure communication in WBAN based on node trust and data trust. Node trust is calculated using direct trust calculation and node behaviours. The data trust is calculated using consistent data success and data aging. The performance is compared with an existing protocol like Trust Evaluation (TE)-WBAN and Body Area Network (BAN)-Trust which is not a cryptographic technique. The protocol is lightweight and has low overhead. The performance is rated best for Throughput, Packet Delivery Ratio, and Minimum delay. With extensive simulation on-off attacks, Selfishness attacks, sleeper attacks, and Message suppression attacks were prevented.

**Keywords:** Wireless Body Area Networks (WBAN); trust; direct trust; security; data trust

## 1. Introduction

The Wireless Body Area Network (WBAN) has become a vital technology with rapid advances in the medical field. However, WBAN faces security issues due to open-air communication of information. Apart from other challenges like energy efficiency, security is a major concern in WBAN in terms of data integrity, data security, and the need to protect from various external and internal attacks. Traditional authentication and encryption technology to protect nodes and data from external attacks are available, but this technology cannot be used in WBAN due to its node constraints [1].

Trust plays an important role in healthcare as it directly influences the quality, adherence, and continuity of the relationship between doctors and patients. Trust is categorized into soft trust and hard trust. There have been models based on hard trust but not enough research related to soft trust even though soft trust plays a great role in dynamic decision-making [2].

The traditional security for WBAN requires highly sophisticated software, hardware, large memory, and high computation and communication capability at the node. At the same time, this technique protects from internal attacks by malicious nodes. Therefore, trust-based security is used to protect the network from internal attacks. Various trust management techniques like Trust Computation, Trust reputation, Trust aggregation, and Trust propagation are there to protect data and nodes from various attacks from the malicious node. The trust computation using calculated with direct trust, the neighbor

node's direct observation, and indirectly recommended by the third node. The calculation of trust reputation is based on the history of communication between the nodes [3].

Wireless Body Area Sensor Networks are used to sense body temperature, heart rates, and other human parameters. Protection and securing of data from various attacks like badmouthing attacks, whitewashing attack, on-off attack, Sybil attacks, ballot stuffing attacks, selective forwarding attacks, Message suppression attacks, selfishness attacks, and sleeper attacks, during internal communication also is a major role.

In this system, we have identified various trust metrics and models used in various technologies like Wireless Sensor Networks, Ad-Hoc Networks, and Wireless Body Area Networks. With existing technology and trust metric and models, a trust model is proposed to identify various malicious behavior with the help of multi-factor trust properties. To identify trusted nodes, we use the direct trust calculation, and using trust value we identify the cooperative and non-cooperative nodes. The second task is to communicate trusted data, the higher the confidence more the trusted data. The consistency of the node success is calculated as well as the freshness of data–more recent data is more trusted data. Summation of all the multi-factor trusted values gives final trust. That trust value is used for secure and quality data communication in WBAN. It detects and avoids attacks during communication, i.e., Message suppression attacks, selfishness attack, and sleeper attack, on off.

The rest of the paper is organized as Section 2 is Literature Review, Section 3 is the Proposed method, Section 4 is the Results and discussion, and Section 5 is the Conclusions.

## 2. Literature Review

This system describes the importance of data quality, with various features and types to satisfy online customers [1]. The paper also proposes a model prediction-based security routing for Wireless Body Area Networks (WBAN). This model increases reliability and prevents data injection attacks [2].

Wireless body area networks (WBAN) are useful for performing cheap, unobtrusive, and constant real-time monitoring of medical applications. Wearable, implantable medical and mobile devices are responsible for controlling critical data, and ensuring security, a critical feature in WBAN. The paper proposes an autonomous mobile agent-based intrusion detection system addressing security in wireless body area networks. Mobile agent-based IDS proposed in other domains with WBAN systems, were compared, and the unique benefits of the proposed architecture were highlighted [3].

In this article, the nodes do not compute and calculate trust values for all the nodes connected in the network. Instead, the nodes calculate the trust values of their neighboring nodes. To calculate the trust, it uses a set of factors such as location identification, nodal distance, sensing and communication, sensing results, consistency, battery level, and trust value. Consistency and sensing communication plays a major role and is used to identify selfishness and save battery power. Selfish nodes do not send data. The consistent level is used to identify malicious nodes, data success, and failure rate, where the range of consistency is $-1 \leq C \leq 1$ [4].

(WBAN) is a collection of wireless sensor nodes that can be placed within or outside the human body monitoring the function and adjoining situations of the body. These devices access private patient medical data, therefore intruding into the subject of privacy protection [5].

This system has designed a framework for WSN based on a beta reputation for trustworthiness, taking aging as one of the factors to identify malicious nodes. Aging measures node's cooperation and also shows the most recent data. Good cooperation shows a good reputation. The range of aging is (0,1) [6].

For secure WSN communication, this system has proposed a multi-angle trust model, for communication trust, data trust, and energy trust. The communication trust is calculated by the Beta probability distribution function. It's used to calculate direct and indirect trust for successful and unsuccessful transactions. Data trust is calculated with the help of fault tolerance, credibility, and consistency [7,8].

The paper mentions monetary problems for healthcare and suggests Body Area Sensor Networks (BASNs) achieve universal healthcare solutions. But for these networks to be properly implemented, they need to be made highly secure and reliable. They present a trust-based aware model along with an entropy-based credibility evaluation model which provides testimony (past evidence-based) about neighbors to ensure security. The reputation of a node is derived from trustworthiness value, derived from aggregated trust pieces of evidence on the body network. This system is said to be viable [9]. In this article, we look into two main factors important for secured data in WBAN, the first being distributed data storage and fine-grained data storage [10].

Wireless Body Area Network, also known as WBAN, is one among many emerging phenomena in the field of medicine currently. Even though it is challenged by security issues for WSNs (Wireless Sensor Networks) and to overcome them. We need to propose a feasible hybrid security mechanism to meet the security requirements of WBANs with strict resource constraints. With the help of these mechanisms, characteristics are analyzed and the main risks of WBANs are and based on risks, the security requirements of WBANs are described and analyzed by available cryptographic algorithms. The proposed mechanism also has a good trade-off between security and resource constraints and provides primitive developed, efficient, and secure WBAN systems, proving beneficial and helpful in the future as well with regular updates and improvements [11].

This system focuses on BAW-based transceivers targeting wireless networks for biomedical applications. BAW resonators are used for frequency synthesis and filtering of the receiver and transmitter paths. The receiver takes advantage of BAW selectivity to perform signal selection and amplification without requiring quadrature on the RF [12].

Energy consumption, a major challenge in WBAN, was minimized using distributed inter-network interference algorithm using game theory [13].

Wireless Body Area Network is known for its context-aware sensing and amplification in sensitivity and specificity. In this system, they have used several tiny wireless sensors placed on a human body. Those wireless sensors provide a methodical monitoring of various vital health signs. As this system collects personal data, the security and privacy of data are of the utmost importance. The difficulty has been faced in implementing security infrastructure on such lightweight protocols. This problem has been approached in such a way that physiological signals (electrocardiogram (ECG)) are exploited to address security issues in WBAN: a Trust Key Management Scheme for Wireless Body Area Networks. This approach manages the generation and distribution of symmetric cryptographic keys [14].

A lightweight authentication scheme for body area networks–BANA, is proposed which does not depend on prior trust among nodes. It is achieved by exploiting characteristics of the physical layer unique to a BAN. Experimental validation of results shows that the clustering method is effective in distinguishing on-body sensors from off-body nodes. Prospects involve the exploration of a more effective solution to thwarting strategic attackers with an overwhelming number and studying the practicality of attacks using directional antenna [15].

In this work, trust-based data is accessed from partners based on certain observations which guarantee the accuracy of the data. The confidence in trust values is calculated based on three factors, history of the previous experience with the neighbor node, age of experience, and the node's good behavior in the initial stage, but later may not cooperate. Thus the recent experience imparts more trust value than the older one. Difference of experience, because the node may behave differently than expected. All this is considered, for trust calculation and confidence in trust data sharing from the partner node [16].

Since almost all healthcare devices work on WSNs which makes these devices are vulnerable to security attacks. Also, there lies a possibility of data manipulation and monitoring. Hence data must be transmitted over a secure network and encrypting it which prevents data leaks or manipulation. Some tools used to accomplish this task are: HealthGear, MobiHealt, ewatch, Ubimon, Code Blue, The Vital Jacket, and many more [17].

For resistant sensor data, this system focuses on data integrity and sensor trust for wireless sensor networks. Sensor trust uses the history of transactions and measures the current risk level to calculate the trust level for communication, robustness, and data integrity. To rate data, the integrity Gaussian model was used. The Gaussian model is used to estimate the accuracy of the data integrity [18].

In this system, various trust metrics were used for node-changing behavior. To identify malicious nodes direct trust, confidence and reputation were utilized. The direct trust is calculated by the delayed Ack method in an organic computing system. Confidence helps to find the reliability of trust values. For that confidence calculation is done in three methods; several confidence and more interactions give high value; age confidence describes how fresh and recent the communication is. So old data reduces confidence. Variance confidence defines the difference in sensing data. The reading very close to the expected value gives more confidence and trust [19].

In this system, the author has designed and analyzed a scalable, adaptive, and survivable trust management protocol for a community of interest-based dynamic social IoT. The protocol is distributed and each node only updates trust towards others of its interest upon encounter or interaction events. Dynamic adaptability is attained by selecting the best trust parameter setting in response to changes to communities of interest. It is also analyzed the trust protocol performance for a newly joined node to the network. Making use of existing trust information in the network, a newly joining node can quickly build up its trust relationship with desirable convergence and accuracy behaviour. Lastly, for scalability, they have proposed a storage management strategy to powerfully use the limited storage in IoT devices. The results show that using the proposed method, this trust protocol with limited storage space achieves a similar performance level as that with ideal (unlimited) storage space and can perform better in the trust convergence time [20].

WBAN monitors the health and transmits data through wireless media. Since wireless media is prone to attacks various key-based cryptography techniques are used, which can be pretty expensive. So this system provides an energy-efficient cluster-based hybrid security framework [21].

SEA-BAN: Semi-Autonomous, was proposed as a vitality effective cluster-based routing protocol for WBANs, named as semi-autonomous adaptive routing in wireless body area network (SEA-BAN). SEA BAN conveys vitality dispersal equitably among body networks and upgrades network lifetime. It is not just a cluster-based routing protocol; it likewise incorporates the advantages of direct transmission and multi-hop transmission techniques. It is contingent upon the vitality level and spatial data of body hubs, to formalize an adaptive routing. In addition, its centralized activity diminishes the computational weight of body networks [22].

Wireless Body Area Network (WBAN) has gained importance as an essential part of medical service due to rapid advancement in medical technology. However, WBAN faces various security issues due to open-air information communication. In this system, the lightweight smart crypto solution was proposed, using authenticated key exchange coupled with cluster head formation and selection for the security of WBAN. The proposed solution logically combines cluster head selection with key agreement fulfilling the security requirement of a wireless body area network, efficient in terms of resource utilization [23].

For data authentication within WBAN, a biometric-based security framework is proposed. Specifically, the sender's electrocardiogram (ECG) is selected as the biometric key for the data authentication mechanism within the WBAN system. Therefore, patients' records can only be sensed and derived personally from the patient's dedicated WBAN system and cannot be mixed with other patients. The security system in WBAN is implemented with low computational complexity and high-power efficiency [24].

This system proposed a Trust Evaluation (TE-model) for Wireless Body Area Network, containing two steps; the first being opinion generation and the second step opinion combination. The calculation of trustworthiness is based on subjective logic. The opinion is generated from sensor attributes, for example, body temperature and ECG values [25].

A special purpose Wireless Sensor Network (WSN) enables remote monitoring termed WBAN. This protocol is used to minimize energy consumption [26].

In this survey, a review of ongoing research in WBANs in terms of system architecture, address allocation, routing, channel modeling, PHY layer, MAC layer, security, and applications is provided [27]. The paper explains that anomaly detection can improve efficiency and reliability [28].

In this paper, we developed and analyzed an adaptive trust management protocol for social IoT systems and its application to service management. Our protocol is distributed and each node only updates trust towards others of its interest upon encounter or interaction events. The effect of $\alpha$ and $\beta$ on the convergence, accuracy, and resiliency properties of our adaptive confidence management protocol through simulation was analyzed. The results demonstrate that the trust evaluation of adaptive trust management will converge and approach ground truth status, and one can tradeoff trust convergence speed for low trust fluctuation, and adaptive trust management is resilient to misbehaving attacks. It verified the effectiveness of adaptive trust management with two real-world social IoT applications. The adaptive trust-based service composition scheme outperforms random service composition and approaches the maximum achievable performance based on ground truth [29]

In this survey, a comparison of various energy-efficient and reliable wireless communication protocols is provided. WBANs are used to enable healthcare professionals to monitor patients and elderly people continuously in their residential environments. A comparison of existing low-power communication technologies potentially supports the rapid development and deployment of WBAN systems [30].

The WBAN is the latest technology used in monitoring health parameters for both normal individuals as well as for patients suffering issues, with privacy protection and security of the collected data of that particular individual. To help authenticate patients' critical data and prevent data corruption during unauthorized access or data erasures during DoS attacks, distributed data access is used alongside other access control methods [31].

In this system, an attack resistance and malicious node detection for BAN is developed based on collaborative filtering called BAN-Trust. The Body sensor data is detected with the detected data sent for data analysis. The data analysis contains two sections; the first one is malicious node detection, the set of malicious attackers considered as a simple attack, where nodes do not provide any necessary information about neighbour nodes. Bad-Mouth Attack nodes spread fake information about neighbour nodes so that malicious nodes will not be detected. On and off attack nodes change the pattern. The second one is trust management. Collaborative filtering can be used to calculate trust to recommend neighbouring nodes using cosine similarity. The recommendation trust contains trust rating formation, trusted neighbour node selection, and prediction trust calculation [32].

Routing protocols are established to decide which path a data packet must travel to reach its destination (sink). In this system, an efficient alternative to the traditional ATTEMPT algorithm has been provided to come up with a routing protocol to save energy. The technique focuses on cluster head selection. If a cluster has not spent much energy and still has energy larger than the threshold level, it will remain cluster head for the next round. In this way, energy is saved rather than wasted in selecting a new cluster head and routing the packet through the new cluster. If the cluster head has lessor energy than the threshold level, it will get replaced by a new cluster head that has larger energy than the threshold level. Thus, saving a lot of energy required to transfer a packet from source to destination [33].

In this system, a trusted communication approach is aimed at protecting the integrity and confidentiality of the WBAN. To ensure authenticity, an improved bilingual Trust Management System (PDATM) is used, whereas a cryptographic system is used to maintain anonymity. An advanced bilingual distribution-based trust-management system (BDTMS) for EH-WBAN is proposed. A MATLAB simulator is used to assess the effectiveness of the recommended program. In the simulation results, BDTMS has a faster detection period

and larger accuracy than the conventional confidence program. Furthermore, BDTMS can defend against inadequate mouth attacks. In medical applications, WBAN allows for continuous monitoring of the patient, allowing for early detection of aberrant conditions and resulting in considerable improvements in quality of life [34].

The WBAN applications and architecture in this article, highlighting current research themes and challenges. As sensor nodes in BANs have limited resources, energy efficiency and reliability are key considerations. We evaluated current advanced energy efficiency and reliability solutions, identifying their strengths and disadvantages. We have also identified several performance features which need to be taken into account when creating and evaluating WBAN solutions. Finally, we identified a number of issues and obstacles in terms of energy efficiency and reliability in WBANs that need to be addressed further in the development of new WBAN solutions. As part of our current study, we introduced a new, efficient and reliable routing strategy to improve the network's stability period and service life in WBAN. In this research work, the emerging technology like edge computing, Fog computing, and AI techniques is not discussed [35].

## 3. Proposed Method

This section defines proposed Trust based data communication for secure WBAN. All body sensor nodes execute successful communication and the trust values calculated for all the nodes to avoid malicious nodes or compromised nodes communicating because the data is communicated within the network is highly sensitive and confidential.

### 3.1. Network Assumption

Assuming Wireless Body Sensors are deployed and formed WBAN, where a set of nodes form a cluster, based on LEACH. All the sensors sense and directly communicate with the centralized node called as WBAN coordinator. The main focus is to develop a secure network and secure and trust-based data communication to the centralized node. This trust-based communication develops resilience from malicious and comprised nodes. Every sensor nodes are aware of the location, and all sensor is capable of sensing event and know where to communicate. Initially equal trust value is assigned to all the sensors deployed in the network.

### 3.2. Trust Computation

For trust computation, two parameters are considered to calculate trust between two nodes. The first one is Node trust, the general metrics which are used to calculate trust are direct trust, Indirect Trust, Reputation, Recommendation, and Cooperation. The second trust metric is Unselfishness, Competence, Accuracy, Significance, Precision, data Volume, aging, centrality, contemporary, and confidence describes the importance of data trust. The second metric is the QoS character for trust calculation. With the above two metrics, Direct Trust and Node Behaviour was used to detect malicious nodes by Node Trust and for secure Communication and trusted data transfer. We are considering consistency and aging as important factors to calculate data trust. The application WBAN is the most sensitive application in healthcare technology. Data needs to be confidential, and the importance of the node and its feature is very critical. So trusted node and trusted data are two important parameters required for trust.

Node $a$ evaluates the trust of neighbour node $b$ $T_{ab}$ at the *time* ($t$), with the range of (0, 1). 1 is the higher trust, 0.5 is the threshold, and 0 is untrusted. For communication and trust formation we assign a summation of all the weights ($W_i$) as equal to 1 where $W_i = W_x + W_y = 1$. All the metrics have equal weightage. The proposed trust model and trust calculation are explained further as follows. $T_{ab}(t)$ is the trust computation between nodes $a$ and $b$ at the time ($t$) [20,29].

$$T_{ab}(t) = W_i T_{ab}^{(Trust\ parameters)}(t)$$

$$T_{ab}(t) \;=\; W_x\, T_{ab}^{Node\ Trust}(t) + W_y T_{ab}^{Data\ Trust}(t)$$

$$Node\ Trust \;=\; Direct\ Trust\ Computation \;+\; Node\ Behaviour$$

$$Data\ Trust \;=\; Data\ Confidence \;=\; Consistent\ and\ Aging$$

$$T_{ab}(t) \;=\; W_x\big(T_{ab}^{Direct\ Trust}(t) + T_{ab}^{Node\ Behaviour}(t)\big) \;+\; W_y\big(T_{ab}^{Consistant} \;+\; T_{ab}^{Aging}(t)\big)$$

where: $-Wx + Wy = 1$

$$T_{ab}(t) \;=\; 0.5\,\big(T_{ab}^{Direct\ Trust}(t) + T_{ab}^{Node\ Behaviour}(t)\big) + 0.5\big(T_{ab}^{Consistant}(t) + T_{ab}^{Aging}(t)\big)$$

### 3.3. Node Trust

For secure internal communication within the Cluster in WBAN, trust-based communication is divided into two sections. The first section identifies the trusted node and avoids malicious nodes. For direct trust, the calculation process is identified and used along with the trusted node behaviour identification.

### 3.3.1. Direct Trust Computation

A Direct Trust DT is calculated using the following equations, lets $T_{ab}^{dt}(t)$ be the direct trust calculated between node $a$ and $b$ at time $t$, calculated using certain parameters. The $DT = T_{ab}^{dt}(t) = range\ of\ (0,\ 1)$, 0 indicates low trust, value 1 indicates high trust, and the threshold value for the Trust is 0.5. Therefore, the communication between any nodes with the value of $T_{ab}^{dt}(t) > 0.5$ is considered trusted communication, where $\Delta t$ is elapsed time, $\alpha T_{ab}^{his}(t)$ considered as a history of trusted communication. We can say accumulated values of previous experience, $T_{ab}^{dt}(t)$ will always hold and calculate new trusted or current trust values with the sum of previous values. $\alpha$ is used to weigh the factor for the trust value, range, (0,1). The equations give the Direct Trust value calculated between node $a$ and node $b$ over time (0,$t$). For our research, only direct trust was considered, by assuming all body sensors directly communicated and connected to the BAN coordinator [20,29].

$$DT = T_{ab}^{dt}(t) = \left[\big(1 - \alpha\, T_{ab}(t - \Delta t) + \alpha T_{ab}^{his}(t)\big)\right]$$

### 3.3.2. Node Behaviour

The calculated Direct Trust value $T_{ab}$ between node a and node b was used to identify node behavior $NB_{ab}$. if the $T_{ab}$ value is greater than the Trust Threshold ($T_h = 0.5$) value, shows the cooperation between the nodes, if it is less than 0.5 shows no cooperation between nodes [6].

$$T_{ab}^{Node\ Behaviour}(t) = NB_{ab} = \begin{cases} cooperate\ \forall\ T_{ab} \geq T_h \\ don't\ cooperate\ \forall\ T_{ab} < T_h \end{cases}$$

### 3.4. Data Trust

The second section is Trusted Data Communication, to identify trusted data consistency of data success by node and aging calculation. This can rely confidently upon continuous observation of data communication. The trust value is increased slowly if the node performs well. The trust value is reduced immediately if it misbehaves.

### 3.4.1. Consistency

The following sections and equations define Trust based data security between nodes in WBAN. Where $T_{ab}^{Nc}$ is defined as a Trust calculation for Node consistency $ds_{ab}$ is defined as data success between nodes a and b, where $df_{ab}$ is defined as data failure between nodes $a$ and $b$. The range is $-1 \leq T_{ab}^{Nc} \leq 1$. Positive values and values close to 1 show more consistent communications and more successful data communication between these nodes. More consistency reduces selfishness attacks. If the node is selfish, it will not participate

in the communication to minimize energy consumption. The Trust value will go negative. With consistent values, malicious and compromised nodes will be identified. We can avoid message suppression attacks [4,8].

$$T_{ab}^{Consistant}(t) = T_{ab}^{Nc} = \frac{ds_{ab} - df_{ab}}{ds_{ab} + df_{ab}}$$

### 3.4.2. Aging

The age of the data generated by a node is calculated by the difference between the current time, $t_c$ and the measurement time of node at time $t$, $t_m$. It shows the quality of data reaching time [6].

$$age_{ab} = t_c - t_m(a)$$

The freshness of data from nodes $a$ to $b$ $U_{ab}$ shows age and current transaction calculated and how fresh data is. The lifetime of the data ranges from (0,1). If $U_{ab}$ value is very less compared to lifetime, which denotes data is very old or outdated data. If the value is close to 1, the data is very fresh and recent. The aging plays a major role in identifying sleeper attacks or off attacks, because nodes perform well, behave in the initial stages and later the same nodes perform as malicious nodes. In such cases, it will not send data to minimize energy consumption and perform sleeper attacks. During this time, the age of the data will be zero, data will simply store in a node, and the lifetime of data reaches zero [6,7,16,19].

$$T_{ab}^{Aging}(t) = U_{ab} = \begin{cases} 1 - \frac{age_a}{Lifetime_{ab}} : if\ age_{ab} < Lifetime_{ab} \\ \qquad\qquad 0 : Other\ wise \end{cases}$$

Final Trust (*FT*) is calculated below,

$$FT = T_{ab}(t) = 0.5\ (T_{ab}^{Direct\ Trust}(t) + T_{ab}^{Node\ Behaviour}(t)) + 0.5(T_{ab}^{Consistant}(t) + T_{ab}^{Aging}(t))$$

## 4. Results and Discussions

The research work is conducted with the help of Omnet++ Simulator, the simulator parameters as given in Table 1.

**Table 1.** Simulation parameters.

| Parameter | Value |
|---|---|
| Sensor Field Region | $(100 \times 100)\ m^2$ |
| Location of Base Station | (50, 150) m |
| Initial Energy of node | 40 J |
| Packet Size | 512 bytes |
| Number of rounds | 50 |
| Number of Nodes | 100 |
| MAC | 802.15.4 |
| Routing | LEACH |
| Simulation Time | 100 s |
| Mobility Model | Fixed |
| Transmit Power | 0.660 W |
| Receiving Power | 0.395 W |
| Transmission range | 50 m |
| Constant Bit rate | 500 kbps |

The performance of the protocol is explained with a graphical representation. The various sensors used in WBAN and their range of the parameter is the Pulse rate sensor. To measure the Oxygen level Body Oxygen level sensor was used. To know the body temperature in the simulation temperature sensor was used = 30–40 °C, ECG = 0.5–5 mV, and EEG = 5–300 μVsensors. Respiratory 2–50 breaths/min, Blood Flow = 1–300 mL/s, Blood Pressure = 0–400 mm Hg.

Figure 1 shows the performance Throughput with increasing malicious nodes for TE (Trust Evaluation) protocol or WBAN, BAN-trust, and Proposed trust model. At the initial level, the percentage of malicious nodes is very less. The performance of the protocol is high, where trust value gradually increases. The identification of the percentage of malicious nodes increased, and the performance of data delivery decreased by the TE-WBAN and BAN Trust when compared to the proposed method.

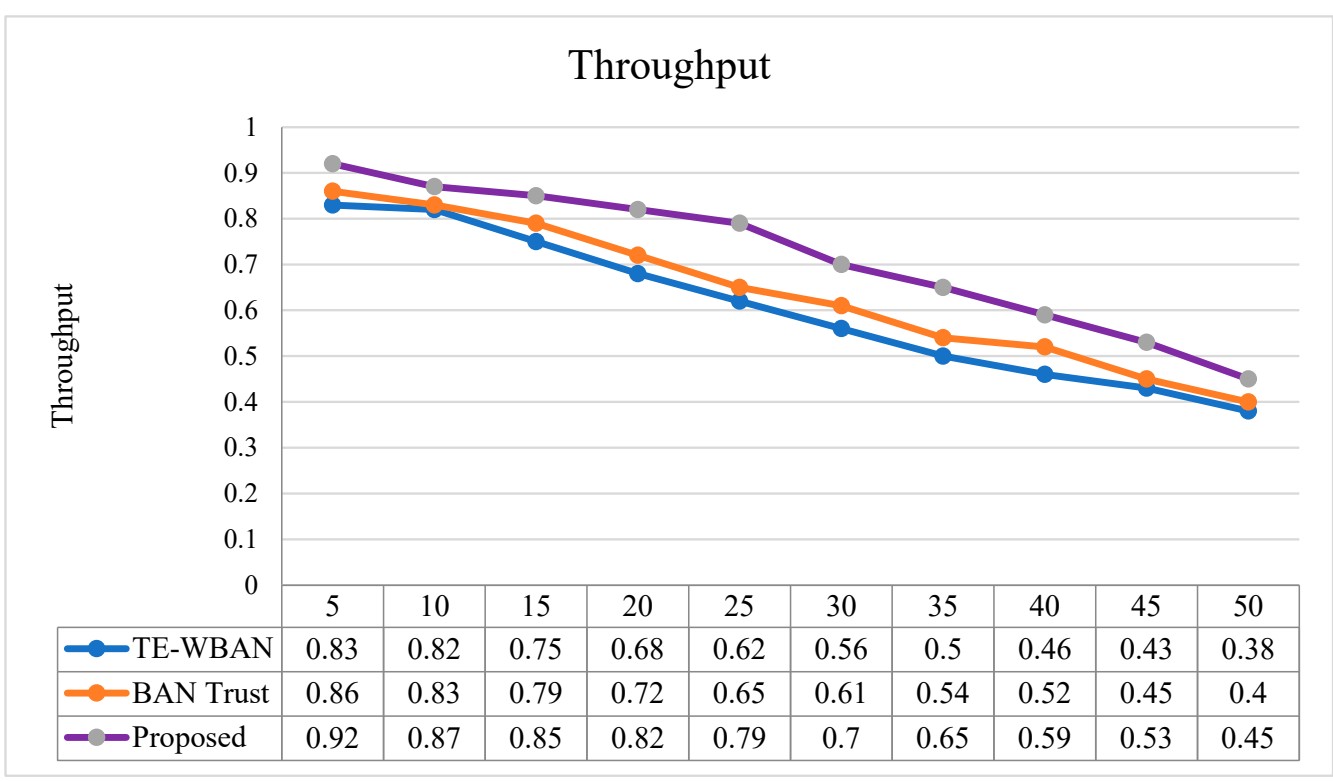

**Figure 1.** Throughput vs. Malicious Node.

The proposed method uses multi factors and packet drops are very less compared to other protocols. This calculates the sum of successful transmission over the channel without any error by the total number of transactions.

Figure 2 shows performance measures of packet delivery ratio, which is the percentage of the message delivered to the total number of messages generated. The performance of the proposed method is more with increasing malicious nodes compared to other protocols like TE-WBAN and BAN-Trust. TE-WBAN and BAN-Trust result in more packet dropped than the proposed trust model. The proposed model detects and avoids message suppression attacks and detects the selfishness nodes as malicious nodes increasing the number of packets delivered when compared with other protocols. The consistency of success rate is more in our model due to multi-factor trust classification, the identification of malicious nodes is faster, leading to a higher packet delivery ratio.

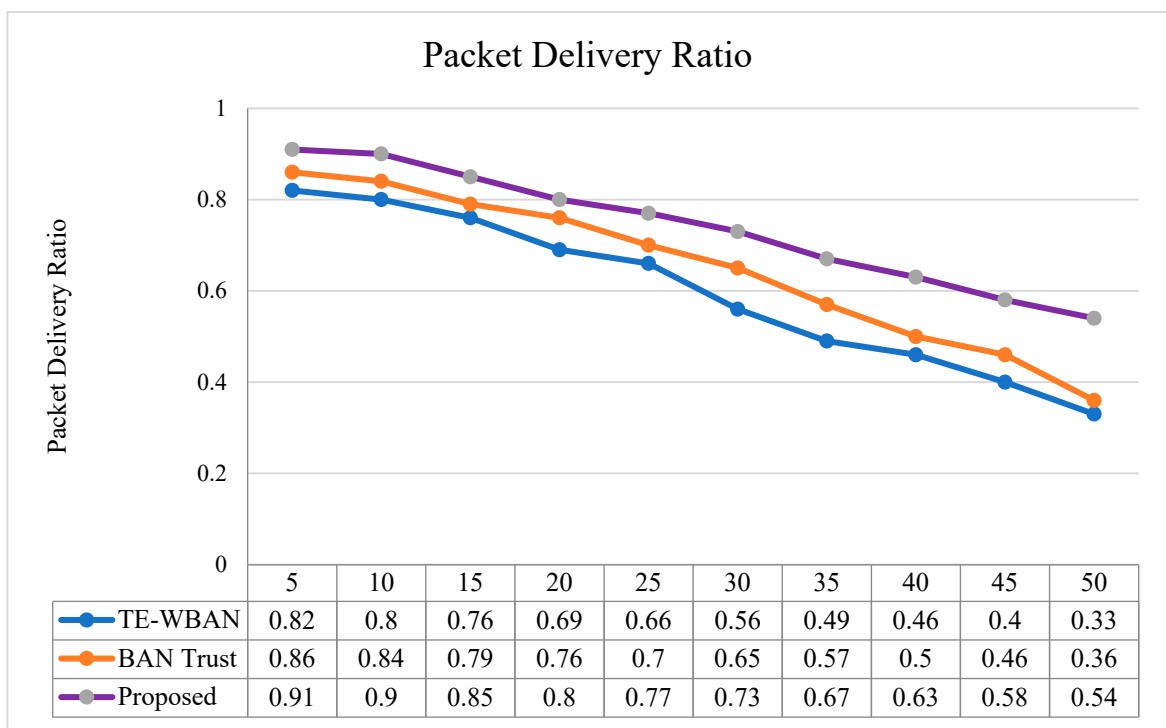

**Figure 2.** Packet Delivery Ratio.

Figure 3 shows comparisons of End to end delay vs. Percentage of malicious nodes with various protocols. The delay is more when malicious nodes are fewer, and gradually reduces and lessens with more malicious nodes. The proposed protocol performs well in (25%to 50%) of malicious nodes compared to TE-WBAN and BAN-Trust. The proposed protocol avoids sleeper attacks and On-and Off attacks throughout the communication process. At all times the cooperative nodes perform well.

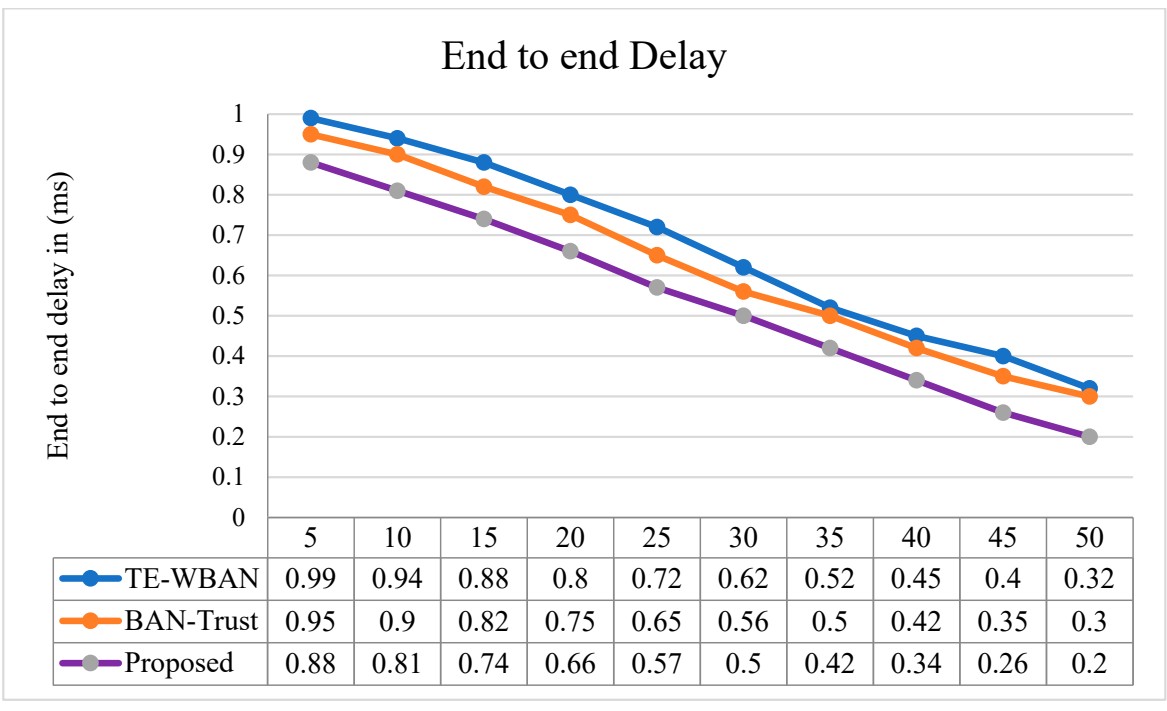

**Figure 3.** End-to-end delay vs. Percentage of malicious nodes.

Figure 4 shows the performance of the protocol's quick response in detecting malicious nodes. All nodes start with the same threshold trust value which is 0.5. During communication with the properties of multifactor, the trust values of nodes in the proposed model are increased gradually concerning the simulation in seconds, compared to TE-WBAN and BAN-Trust. At the same time, as soon as the node turns into the malicious node, trust values are drastically reduced immediately to less than 0.5 in the proposed trust model. As per the result in Figure 4, the proposed model detected malicious nodes between 4 to 5 s, and immediately reduced its trust value to 0.3 at the 5th s, which is an untrusted range, further reducing to 0. But the BAN-Trust and TE-WBAN are detected at the 6th and 7th s respectively. The result shows Proposed Trust model detects malicious behavior faster than the other two protocols.

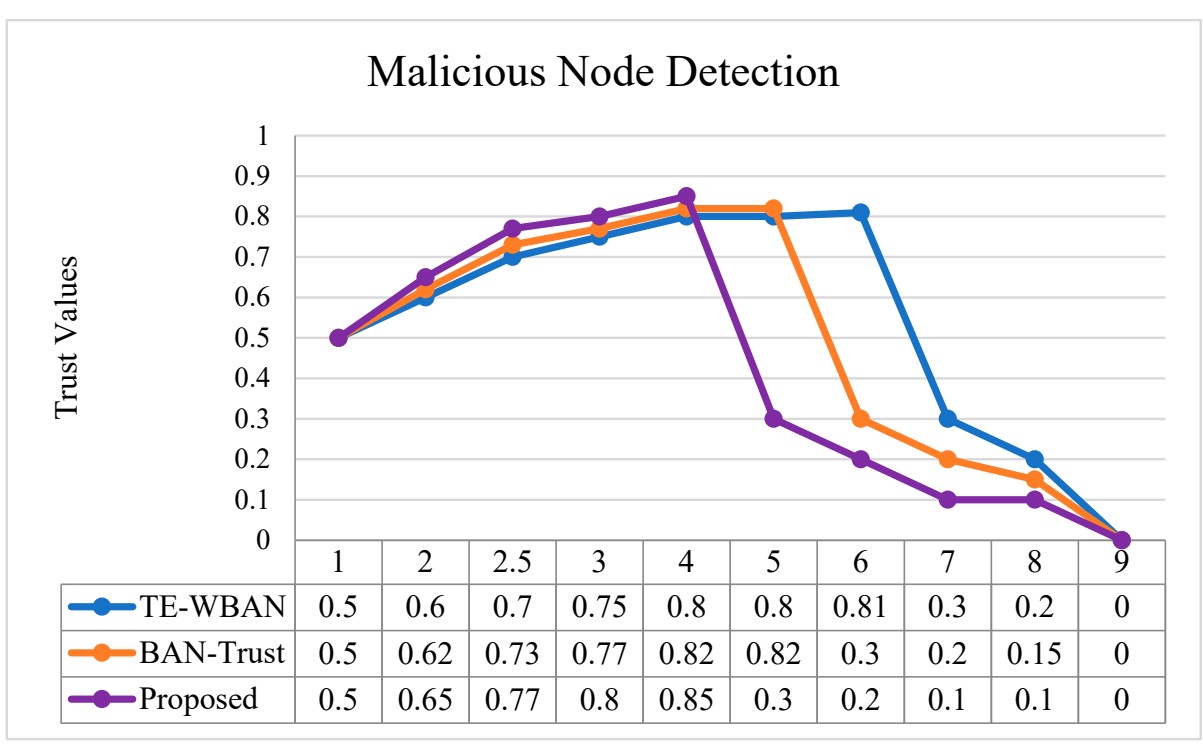

**Figure 4.** Malicious Node Detection.

### 5. Conclusions

In this system, various existing trust management schemes for secure data communication protocols to avoid internal soft attacks on WSN are discussed and the research gaps are identified. It is found that there is not much contribution in trust management for WBAN. The trust parameters and metrics chosen in WSN are also suited for security in WBAN. We proposed a suitable trust model for secure communication in WBAN based on node trust and data trust. Node trust is calculated with direct trust calculation and node behaviors. The data trust is calculated by the consistency of data success and aging or the freshness of the data, which gives confidence in the sensed data. We proposed Trust Model providing secure data transmission and the performance is compared with the existing protocol like TE-WBAN and BAN-Trust, which is not a cryptographic technic, so the protocol is lightweight and has low overhead. The performance is best for throughput, Packet Delivery Ratio, and minimum delay. The malicious behavior identification by the proposed trust model is fast compared to TE-WBAN and BAN-Trust. All these parameters are compared with varying malicious node percentages. At the same time, On-Off attacks, Selfishness attacks, sleeper attacks, and Message suppression attacks were prevented.

**Author Contributions:** Both Authors have an equal contribution. S.R. is research scholar and U.D.G. is corresponding author and research supervisor. All authors have read and agreed to the published version of the manuscript.

**Funding:** There is no external research funding.

**Data Availability Statement:** All are simulated data, it's a dynamic data generated during simulation.

**Conflicts of Interest:** The authors declare there is no conflict of interest.

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
