# Peer review of "Trust-Based Data Communication in Wireless Body Area Network for Healthcare Applications"

_2504-2289, doi:10.3390/bdcc6040148_

Round 1
Reviewer 1 Report
Explain in detail the proposed methodology. Overall writing and presentation needs improvement. Why the authors claim this system for Heathcare Applications?. Authors in Section 4 claiming that this is carried out in Omnet++ Simulater. Why Figure 1, Figure 2 and Figure 3 were plotted using Excel. More results needed for supporting proposed system.
Lots of typos and grammar mistakes (Figure 2). Seems like the results are manipulated. What actually is the proposed system?. How the proposed system provides better results?. The information highlighted is already known and lots of advanced papers are published in this regard.
Author Response
Dear sir ,
thank you for your valuable comments.

Reviewer 2 Report
1.Introduction section needs to be re-written to improve its quality and readability.
2.What is the motivation of the proposed work? Research gaps, objectives of the proposed work should be clearly justified
3.Overall, the basic background is not introduced well, where the notations are not illustrated much clear.
4.The literature has to be strongly updated with some relevant and recent papers focused on the fields dealt with in the manuscript.
5.Proposed work can be summarized in a figure
6. Explain why the current method was selected for the study, its importance and compare with traditional methods.
7.Authors are suggested to include more discussion on the results and also include some explanation regarding the justification to support why the proposed method is better in comparison towards other methods
8.The language usage throughout this paper need to be improved, the author should do some proofreading on it.
9. References 4 to 34, the numbers are repeated.
10. Check Journal template
11.How this proposed system works for healthcare applications? No details were found throughout the manuscript.
12. What really Figure 1 2 and 3 stand for? Where these figures come from? Where are supporting data related to this figures? Seems like the figures are generated using MS word.
13. Authenticity of generated results needs proof
Author Response
Dear sir,
thank you for your valuable comments.

Reviewer 3 Report
1. Avoid using the word "this paper" instead use system/model/method.
2. Few related works are very less [20] [26], can be explained more.
3. Table 1 is not clear
4. Explain how trust model is used in detecting the attacks.
5. Recent references need to be included.
6.simulation screenshots can be included and some more explanation required.
Author Response

(The authors gave the same response as above.)

Reviewer 4 Report
The manuscript is interesting and useful. it can be accepted after revision. I have two main concerns and these need to be addressed before final acceptance.
1. Authors need to add more detail focused discussions on Figure 1, 2, 3, and 4.
2. Secondly authors need to add bench mark table by giving the comparison of current study with literature. How the model behave different from other models and how it is better.
Author Response

(The authors gave the same response as above.)

Round 2
Reviewer 1 Report
authors have revised all my comments.
Reviewer 2 Report
All the comments are addressed. Thank you.